SciPost Physics

Submission

# Bose polaron in a quantum fluid of light

A. Vashisht[1*], M. Richard[2], A. Minguzzi[1]

**1** Univ. Grenoble Alpes, CNRS, LPMMC, 38000 Grenoble, France
**2** Univ. Grenoble Alpes, CNRS, Grenoble INP, Institut Néel , 38000 Grenoble, France
* amit.vashisht@lpmmc.cnrs.fr

October 14, 2021

## Abstract

We study the Bose polaron problem in a nonequilibrium setting, by considering an impurity embedded in a quantum fluid of light realized by exciton-polaritons in a microcavity, subject to a coherent drive and dissipation on account of pump and cavity losses. We obtain the polaron effective mass, the drag force acting on the impurity, and determine polaron trajectories at a semiclassical level. We find different dynamical regimes, originating from the unique features of the excitation spectrum of driven-dissipative polariton fluids, in particular a non-trivial regime of acceleration against the flow. Our work promotes the study of impurity dynamics as an alternative testbed for probing superfluidity in quantum fluids of light.

# 1 Introduction

In vacuum, light can exchange momentum and energy with free moving particles via the mechanism of radiation pressure. When a photon impinges a particle (e.g. an atom, a glass bead, an interplanetary probe) it can be absorbed, refracted and/or scattered, thereby transferring a fraction of its momentum and energy to the particle, which is thus accelerated and steered by light. This mechanism is at the basis of several key applications and phenomena such as ultracold atom cooling [1,2], cavity quantum optomechanics [3], as well as in proposed interplanetary probes design [4].

This picture is expected to be profoundly modified and enriched for photons propagating in solids, and all the more so for photons strongly dressed with electronic excitations, such as in semiconductor microcavities in the strong coupling regime between excitons (bound electron hole pair excitation) and cavity photons [5–8]. In this case, the dressed photons are called excitons-polaritons (polaritons) [9], and they are characterized by two-body interactions and quantum many-body effects [9] such as collective excitations [10–12].

Experimentally, the polariton quantum fluid [13,14] can be generated in a well-defined momentum state and sent to collide against a fixed obstacle (e.g. a localized barrier potential). This case has been examined both theoretically [15–18] and experimentally [19,20]: it was shown that for a suitable set of experimental parameters the flow around the obstacle exhibits strong signatures of superfluidity [21], evidenced by frictionless flow and absence of drag force, while for other experimental parameters, the drag force is restored [22], with a non-trivial magnitude and directions [23]. In particular, as peculiar in driven-dissipative quantum fluids, a regime of negative drag force was reported to occur when the dispersion relation displays a dispersive imaginary part [22]. In such a regime, due to momentum-dependent dissipation, the forward scattering is more suppressed than the backward one and the fluid effectively feels a force in the direction opposite to the motion of the obstacle.

In this work, we consider the case of a freely-moving finite-mass obstacle subject to the drag force exerted by a driven-dissipative quantum fluid of polaritons. This problem can be understood as the analogue of determining the dynamics of a particle subjected to radiation pressure, when radiation pressure is replaced by the polaritonic quantum fluid drag force. This is also a different class of problem than the fixed obstacle case mentioned above: indeed, a fixed obstacle has an infinite mass and cannot be excited by the fluid, and thus has no dynamics. From a theoretical point of view, the situation that we consider is that of a single impurity dressed by the excitations of its many-body environment, known as the polaron problem. This problem is of high importance in several contexts. In solid-state physics, a polaron state describes electrons dressed by lattice phonons [24–29]. The Fermi polaron, describing a mobile impurity in a degenerate Fermi gas has attracted renewed attention in ultracold atomic gases [30–43] and has also been recently observed in monolayer semiconductors [44,45] when exciton-polaritons propagate in a fluid of free electrons [46,47]. The Bose polaron problem, i.e. the case of an impurity in a bosonic quantum fluid has also attracted significant attention [48–62] and was experimentally demonstrated in ultracold atoms [63–68]. For Bose polaron-polaritons at equilibrium, Feshbach-mediated interactions [69] and formation of photon bound states [70] were predicted. Bose Polaron-

polaritons were also proposed as a probe to observe correlated many-body states [71].

Here, we extend the description of Bose polarons to the non-equilibrium case of a bosonic driven-dissipative quantum fluid, with specific focus on exciton-polaritons. Focusing on the experimentally relevant condition of weak coupling between the impurity and the fluid, we develop a Bogoliubov-Fröhlich approach in the presence of an external bath and provide solutions, under Markovian approximation, for the effective mass of the polaron and the drag force exerted by the fluid on it. We consider both the cases of the fluid at rest and in motion. We then examine the impurity motion induced by the drag of fluid and identify the signatures of superfluidity in polaron dynamics.

From an experimental point of view, a free moving obstacle (or impurity) of finite mass has not been realized so far and yet, realistic strategies can be envisaged. For example, a long-lifetime dark exciton droplet can be created by a second laser beam in the middle of the polariton flow. Such excitons can be dark (i.e. not in the strong coupling regime) due to the symmetry of their wavefunction in multiple-quantum-well microcavities [72], or can have momenta outside the light cone, as a result of thermalization. Another possible impurity, much lighter in mass, can be made up of a cross-polarized polariton droplet created by a second laser. While such polaritons have lower interaction with cross-polarized polaritons, it is still nonzero [73] and could be enhanced by utilizing Feshbach resonance with the biexciton [74].

The paper is organized as follows. We begin with the description of the model for our system in Sec. 2. We then derive the equations for quantum dynamics of the impurity in Sec. 3, allowing us to obtain expressions for the effective mass, drag force and trajectories. The next section 4 is devoted to present our main results, in particular on the drag force, which can either be positive or negative depending on the parameter regimes. We also derive the semiclassical trajectories of the impurity, and discuss the effect of drive and dissipation on the impurity dynamics. We show in particular how it can be used as microscopic probe of the superfluid properties of the polariton fluid. We finally present our main conclusions and outlook in Sec. 5.

## 2   Physical system and model Hamiltonian

We consider an exciton-polariton fluid (called polariton onwards for brevity) coherently driven by a resonant pump and interacting with dilute impurities. Polaritons are hybrid light-matter quasiparticles originating from the strong coupling between excitons and photons (see e.g. [9]). The polariton modes come in two branches: the lower polariton and upper polariton. Here we restrict only to the lower polariton branch neglecting the upper branch since it is detuned far out of the relevant energy scales of our system. A schematic of the system setup considered in present work is shown in Figure 1. We describe the polariton losses through the mirrors by their coupling to the continuum of photons outside the cavity, which thus acts as a bath. The total Hamiltonian of the system reads

$$\hat{\mathcal{H}} = \hat{\mathcal{H}}_{\mathrm{P}} + \hat{\mathcal{H}}_{\mathrm{I}} + \hat{\mathcal{H}}_{\mathrm{B}}. \tag{1}$$

Here $\hat{\mathcal{H}}_{\mathrm{P}}$ describes the polariton fluid coherently driven by a continuous wave laser pump of frequency $\omega_p$ and wavevector $\mathbf{k}_p$ [9] and is given as

$$
\hat{\mathcal{H}}_{\mathrm{P}} = \sum_{\mathbf{k}} \left( \frac{\hbar^2 \mathbf{k}^2}{2m} - \hbar\omega_p \right) \hat{a}_{\mathbf{k}}^{\dagger} \hat{a}_{\mathbf{k}} + \frac{g}{2A} \sum_{\mathbf{k},\mathbf{k}',\mathbf{q}} \hat{a}_{\mathbf{k}+\mathbf{q}}^{\dagger} \hat{a}_{\mathbf{k}'-\mathbf{q}}^{\dagger} \hat{a}_{\mathbf{k}'} \hat{a}_{\mathbf{k}}
$$
$$
+ \hat{a}_{\mathbf{k}_p} F_0^* + \hat{a}_{\mathbf{k}_p}^{\dagger} F_0, \tag{2}
$$

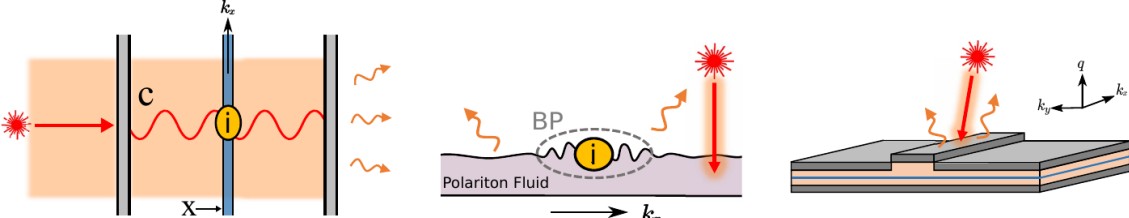

Figure 1: **Schematic of the system setup** considered in the present work. Left panel: excitons (x) and impurities (i) are placed in a photonic cavity (c), pumped by laser light and subjected to losses. Central panel: as a result of the strong light-matter coupling, a polariton fluid is created and the impurity is dressed by its excitations, giving rise to a Bose polaron (BP). Right panel: illustration on how this set up could be experimentally realized in semiconductor microcavities. To achieve the quasi-one-dimensional geometry considered in the present work, a specific etched structure can be realized. The figure shows also how to set into motion the polariton fluid by suitably choosing the angle of incidence of the laser beam.

where $\hat{a}_{\mathbf{k}}^{\dagger}$, $\hat{a}_{\mathbf{k}}$ are the creation and annihilation operators of polaritons, $g$ is the polariton-polariton interaction constant, $A$ is the pumped area in the plane of the microcavity and $F_0$ is the plane-wave incident laser field with wavevector $\mathbf{k}_p$.

Since we consider dilute impurites, we neglect any interactions among them and hence can limit ourselves to the case of a single impurity. $\hat{\mathcal{H}}_I$ describes the impurity weakly interacting with the polariton fluid, and is given as

$$\hat{\mathcal{H}}_I = \frac{\hat{\mathbf{p}}^2}{2M} + \frac{g_{\mathrm{IB}}}{A} \sum_{\mathbf{k},\mathbf{k}'} e^{i(\mathbf{k}-\mathbf{k}')\cdot\hat{\mathbf{x}}} \hat{a}_{\mathbf{k}'}^{\dagger} \hat{a}_{\mathbf{k}}, \tag{3}$$

where $M$ is the mass of the impurity and the second term describes the impurity-polariton interaction, characterized by coupling constant $g_{\mathrm{IB}}$.

Finally, $\hat{\mathcal{H}}_{\mathrm{B}}$ describes the external radiation bath as harmonic excitations, linearly interacting - in the spirit of Caldeira-Legget model [75–77] - with the polariton fluid:

$$\hat{\mathcal{H}}_{\mathrm{B}} = \int dq \sum_{\mathbf{k}} \hbar\omega_{q,\mathbf{k}} \, \hat{\alpha}_{q,\mathbf{k}}^{\dagger} \alpha_{q,\mathbf{k}} + \int dq \sum_{\mathbf{k}} \left[ \kappa_{q,\mathbf{k}}^{*} \, \hat{\alpha}_{q,\mathbf{k}}^{\dagger} \hat{a}_{\mathbf{k}} + \kappa_{q,\mathbf{k}} \, \hat{a}_{\mathbf{k}}^{\dagger} \hat{\alpha}_{q,\mathbf{k}} \right], \tag{4}$$

where $\omega_{q,\mathbf{k}}$ is frequency of the bath mode with in-plane wavevector $\mathbf{k}$ and a continuous wavevector $q$ in the orthogonal direction; $\alpha_{q,\mathbf{k}}^{\dagger}$, $\alpha_{q,\mathbf{k}}$ are the creation and annihilation operators of the bosonic bath modes and $\kappa_{q,\mathbf{k}}$ quantifies the coupling of the polariton modes with the external bath. We neglect the coupling of the impurity with the bath for simplicity.

In the next two subsections, we simplify this Hamiltonian; first by assuming a weak excitation density on top of the condensate at all times, and thus taking the Bogoliubov approximation, and then by moving to the impurity reference frame using the Lee, Low and Pines (LLP) transformation [26].

## 2.1 Bogoliubov approximation

We will consider the three subsystems separately, starting with the polariton Hamiltonian in equation (2), which, following a standard convention, can be conveniently rewritten in

terms of the excitation wavevectors $\mathbf{k}$ shifted with respect to the pump wavevector $\mathbf{k}_p$:

$$\hat{\mathcal{H}}_{\mathrm{P}} = \sum_{\mathbf{k}} \left( \frac{\hbar^2(\mathbf{k}_p + \mathbf{k})^2}{2m} - \hbar\omega_p \right) \hat{a}^\dagger_{\mathbf{k}_p+\mathbf{k}} \hat{a}_{\mathbf{k}_p+\mathbf{k}}$$
$$+ \frac{g}{2A} \sum_{\mathbf{k},\mathbf{k}',\mathbf{q}} \hat{a}^\dagger_{\mathbf{k}_p+\mathbf{k}+\mathbf{q}} \hat{a}^\dagger_{\mathbf{k}_p+\mathbf{k}'-\mathbf{q}} \hat{a}_{\mathbf{k}_p+\mathbf{k}'} \hat{a}_{\mathbf{k}_p+\mathbf{k}}$$
$$+ \hat{a}_{\mathbf{k}_p} F_0^* + \hat{a}^\dagger_{\mathbf{k}_p} F_0. \tag{5}$$

We now use the Bogoliubov approximation - which assumes a macroscopic occupation of $\mathbf{k}_p$ mode $\hat{a}_{\mathbf{k}_p}(\hat{a}^\dagger_{\mathbf{k}_p}) \approx \sqrt{N_0}$ - in (5) to approximate it as a quadratic Hamiltonian in the creation and annihilation operators. It is then diagonalized with the help of Bogoliubov transformation (see Appendix A for details)

$$\hat{a}_{\mathbf{k}_p+\mathbf{k}} = u_{\mathbf{k}} \hat{b}_{\mathbf{k}_p+\mathbf{k}} - v^*_{-\mathbf{k}} \hat{b}^\dagger_{\mathbf{k}_p-\mathbf{k}} \tag{6a}$$

$$\hat{a}^\dagger_{\mathbf{k}_p-\mathbf{k}} = -v_{\mathbf{k}} \hat{b}_{\mathbf{k}_p+\mathbf{k}} + u^*_{-\mathbf{k}} \hat{b}^\dagger_{\mathbf{k}_p-\mathbf{k}}. \tag{6b}$$

The polariton Hamiltonian in the Bogoliubov approximation then reads

$$\hat{\mathcal{H}}_{\mathrm{P}} = E_{ss} + \sum_{\mathbf{k}\neq 0} E_{\mathrm{b}} \hat{b}^\dagger_{\mathbf{k}_p+\mathbf{k}} \hat{b}_{\mathbf{k}_p+\mathbf{k}}, \tag{7}$$

where $E_{ss}$ is the energy of the macroscopically occupied steady state at $\mathbf{k}_p$; it provides a constant energy shift and since it doesn't contribute to the dynamics of the problem we can safely neglect this term moving forward. $E_{\mathrm{b}}$ is the Bogoliubov spectrum of polaritons and is given as

$$E_{\mathrm{b}} = \frac{\hbar^2 \mathbf{k} \cdot \mathbf{k}_p}{m} + \hbar\omega_{\mathbf{k}} \tag{8}$$

where $\hbar\omega_{\mathbf{k}} = \left[ \epsilon_{\mathbf{k}} \left( \epsilon_{\mathbf{k}} + 2gn \right) \right]^{\frac{1}{2}}$, $\epsilon_{\mathbf{k}} = \frac{\hbar^2 \mathbf{k}^2}{2m} - \Delta$ and $\Delta = \hbar\omega_p - \frac{\hbar^2 \mathbf{k}_p^2}{2m} - gn$, with $gn$ the blueshift due to polariton-polariton interactions and $n = N_0/A$ the density of polariton condensate. The parameters of Bogoliubov transformation are given by

$$u_{\mathbf{k}}, v_{\mathbf{k}} = \left[ \frac{1}{2} \left( \frac{\epsilon_{\mathbf{k}} + gn}{\hbar\omega_k} \pm 1 \right) \right]^{\frac{1}{2}}. \tag{9}$$

Note that if $\Delta \leq 0$ we have $u_{\mathbf{k}}(v_{\mathbf{k}}) = u_{-\mathbf{k}}(v_{-\mathbf{k}}) = u^*_{\mathbf{k}}(v^*_{\mathbf{k}})$ but this is not true for $\Delta > 0$ where $u_{\mathbf{k}}(v_{\mathbf{k}}) \neq u^*_{\mathbf{k}}(v^*_{\mathbf{k}})$ and it calls for a special diagonalization treatment which is detailed in Appendix A. For the purpose of legibility, we will work with the former case and use $u_{\mathbf{k}}(v_{\mathbf{k}})$ for all the the instances of $u_{-\mathbf{k}}(v_{-\mathbf{k}})$ and $u^*_{\mathbf{k}}(v^*_{\mathbf{k}})$ in all subsequent derivations. The extension to the case $\Delta > 0$ can be easily made.

The nature of excitation spectrum at low momenta is controlled by the detuning $\Delta$: depending on its value, the excitation spectrum can be gapless, gapped, or exhibit a non-dispersive region at small wavevectors (see panel b in Fig.2).

Using the Bogoliubov approximation in the impurity Hamiltonian (3) and neglecting the term describing the interaction between two Bogoliubov excitations of the condensate which are of order $1/\sqrt{N_0}$ smaller than the interaction of an excitation with the impurity, we obtain

$$\hat{\mathcal{H}}_I = \frac{\hat{\mathbf{p}}^2}{2M} + g_{\mathrm{IB}}n + \sum_{\mathbf{k}\neq 0} e^{i\mathbf{k}\cdot\hat{\mathbf{x}}} \left( V_{\mathbf{k}} \hat{b}_{\mathbf{k}_p+\mathbf{k}} + V_{\mathbf{k}} \hat{b}^\dagger_{\mathbf{k}_p-\mathbf{k}} \right), \tag{10}$$

where $V_{\mathbf{k}}$ is the scattering amplitude of Bogoliubov excitation with the impurity, given as

$$V_{\mathbf{k}} = g_{\mathrm{IB}} \frac{\sqrt{N_0}}{A} \left( u_{\mathbf{k}} - v_{\mathbf{k}} \right). \tag{11}$$

Notice that $V_{\mathbf{k}}$ is associated to the polariton fluid density fluctuations through the term $(u_{\mathbf{k}} - v_{\mathbf{k}})$. In the case of a gapless phononic dispersion, this term is suppressed at low momenta as phase fluctuations dominate the Bogoliubov excitation spectrum. Our analysis hence shows that the impurity dynamics can probe superfluid properties of the quantum fluid of light.

We finally apply the Bogoliubov transformation to the bath Hamiltonian (4), which results in

$$\hat{\mathcal{H}}_{\mathrm{B}} = \int dq \sum_{\mathbf{k}} \hbar \omega_{q,\mathbf{k}} \, \hat{\alpha}_{q,\mathbf{k}}^{\dagger} \alpha_{q,\mathbf{k}} + \int dq \sum_{\mathbf{k}} \left[ \kappa_{q,\mathbf{k}}^{*} \, \hat{\alpha}_{q,\mathbf{k}}^{\dagger} \left( u_{\mathbf{k}} \, \hat{b}_{\mathbf{k}_p + \mathbf{k}} - v_{\mathbf{k}} \, \hat{b}_{\mathbf{k}_p - \mathbf{k}}^{\dagger} \right) \right.$$
$$\left. + \kappa_{q,\mathbf{k}} \, \hat{\alpha}_{q,\mathbf{k}} \left( u_{\mathbf{k}} \, \hat{b}_{\mathbf{k}_p + \mathbf{k}}^{\dagger} - v_{\mathbf{k}} \, \hat{b}_{\mathbf{k}_p - \mathbf{k}} \right) \right]. \tag{12}$$

The sum of equations (7), (10) and (12) provides an approximate description of our system, that takes the form of a Bogoliubov-Fröhlich Hamiltonian [48, 50, 55] with a dissipative bath term.

## 2.2   Lee Low Pines transformation

In order to further simplify our model we use the Lee Low Pines (LLP) transformation [26] to remove the impurity degree of freedom in the impurity-polariton interaction term. This is achieved by applying the operator

$$T_{\mathrm{LLP}} = \exp \left[ -i \left( \sum_{\mathbf{k} \neq 0} \hbar \mathbf{k} \, \hat{b}_{\mathbf{k}_p + \mathbf{k}}^{\dagger} \hat{b}_{\mathbf{k}_p + \mathbf{k}} \right) \cdot \hat{\mathbf{x}} \right] \tag{13}$$

on the full system Hamiltonian, to move to the impurity reference frame. Denoting the momentum of the Bogoliubov excitations as $\hat{\Pi} = \sum_{\mathbf{k} \neq 0} \hbar \mathbf{k} \, \hat{b}_{\mathbf{k}_p + \mathbf{k}}^{\dagger} \hat{b}_{\mathbf{k}_p + \mathbf{k}}$, the resulting total Hamiltonian reads

$$\hat{\mathcal{H}}_{\mathrm{LLP}} = \sum_{\mathbf{k} \neq 0} E_{\mathrm{b}} \, \hat{b}_{\mathbf{k}_p + \mathbf{k}}^{\dagger} \hat{b}_{\mathbf{k}_p + \mathbf{k}} + \frac{1}{2M} \left( \hat{\mathbf{p}} - \hat{\Pi} \right)^2$$
$$+ \sum_{\mathbf{k} \neq 0} \left( V_{\mathbf{k}} \, \hat{b}_{\mathbf{k}_p + \mathbf{k}} + V_{\mathbf{k}} \, \hat{b}_{\mathbf{k}_p - \mathbf{k}}^{\dagger} \right) + \int dq \sum_{\mathbf{k}} \hbar \omega_{q,\mathbf{k}} \, \hat{\alpha}_{q,\mathbf{k}}^{\dagger} \alpha_{q,\mathbf{k}}$$
$$+ \int dq \sum_{\mathbf{k}} \left[ \kappa_{q,\mathbf{k}}^{*} \, \hat{\alpha}_{q,\mathbf{k}}^{\dagger} \left( u_{\mathbf{k}} \, \hat{b}_{\mathbf{k}_p + \mathbf{k}} - v_{\mathbf{k}} \, \hat{b}_{\mathbf{k}_p - \mathbf{k}}^{\dagger} \right) e^{-i \mathbf{k} \cdot \hat{\mathbf{x}}} \right.$$
$$\left. + \kappa_{q,\mathbf{k}} \, \hat{\alpha}_{q,\mathbf{k}} \left( u_{\mathbf{k}} \, \hat{b}_{\mathbf{k}_p + \mathbf{k}}^{\dagger} - v_{\mathbf{k}} \, \hat{b}_{\mathbf{k}_p - \mathbf{k}} \right) e^{i \mathbf{k} \cdot \hat{\mathbf{x}}} \right]. \tag{14}$$

# 3   Quantum dynamical equations and observables

In order to obtain the impurity dynamics, we derive the Heisenberg equation of motion for $\hat{\mathbf{x}}$, $\hat{\mathbf{p}}$, $\hat{b}_{\mathbf{k}_p + \mathbf{k}}$ and $\hat{\alpha}_{q,\mathbf{k}}$ operators. The resulting equations of motion read:

$$\frac{d\hat{\mathbf{x}}}{dt} = \frac{\left( \hat{\mathbf{p}} - \hat{\Pi} \right)}{M}, \tag{15}$$

$$\frac{d\hat{\mathbf{p}}}{dt} = \sum_{\mathbf{k}} i\mathbf{k}e^{-i\mathbf{k}\cdot\hat{\mathbf{x}}} \int dq \Big[ \kappa_{q,\mathbf{k}}^* \, \hat{\alpha}_{q,\mathbf{k}}^\dagger \left( u_{\mathbf{k}} \, \hat{b}_{\mathbf{k}_p+\mathbf{k}} - v_{\mathbf{k}} \, \hat{b}_{\mathbf{k}_p-\mathbf{k}}^\dagger \right)$$

$$+ \kappa_{q,-\mathbf{k}} \, \hat{\alpha}_{q,-\mathbf{k}} \left( u_{-\mathbf{k}} \, \hat{b}_{\mathbf{k}_p-\mathbf{k}}^\dagger - v_{\mathbf{k}} \, \hat{b}_{\mathbf{k}_p+\mathbf{k}} \right) \Big], \tag{16}$$

$$i\hbar\frac{d}{dt}\hat{b}_{\mathbf{k}_p+\mathbf{k}} = \left( E_{\mathrm{b}} + \frac{\hbar^2\mathbf{k}^2}{2M} + \frac{\hbar\mathbf{k}\cdot\hat{\Pi}}{M} - \frac{\hbar\mathbf{k}.\hat{\mathbf{p}}}{M} \right) \hat{b}_{\mathbf{k}_p+\mathbf{k}} + V_{\mathbf{k}}$$

$$+ \int dq \left( \kappa_{q,\mathbf{k}} \, \hat{\alpha}_{q,\mathbf{k}} \, u_{\mathbf{k}} - \kappa_{q,\mathbf{k}}^* \, \hat{\alpha}_{q,\mathbf{k}}^\dagger \, v_{\mathbf{k}} \right) e^{i\mathbf{k}\cdot\hat{\mathbf{x}}}, \tag{17}$$

$$i\hbar\frac{d}{dt}\hat{\alpha}_{q,\mathbf{k}} = \hbar\omega_{q,\mathbf{k}} \, \alpha_{q,\mathbf{k}} + \kappa_{q,\mathbf{k}}^* \left( u_{\mathbf{k}} \, \hat{b}_{\mathbf{k}_p+\mathbf{k}} - v_{\mathbf{k}} \, \hat{b}_{\mathbf{k}_p-\mathbf{k}}^\dagger \right) e^{-i\mathbf{k}\cdot\hat{\mathbf{x}}}. \tag{18}$$

Notice that the impurity momentum is not conserved due to the presence of the bath. Now we proceed to obtain a solution for the four coupled equation of motions. We derive a self-consistent equation for $\Pi$ in the next subsection, using which we can calculate the trajectory of polarons.

## 3.1 Dynamics of the fluid excitations

We commence by tracing out the bath degrees of freedom by substituting the solution of (18) for $\hat{\alpha}_{q,\mathbf{k}}$ in (17). The general solution of (18) is given as

$$\hat{\alpha}_{q,\mathbf{k}} = e^{-i\omega_{q,\mathbf{k}}(t-t_0)}\alpha_{q,\mathbf{k}}(t_0) - i\kappa_{q,\mathbf{k}}^* \int_{t_0}^t dt' \, e^{-i\omega_{q,\mathbf{k}}(t-t')} \left( u_{\mathbf{k}} \, \hat{b}_{\mathbf{k}_p+\mathbf{k}}(t') - v_{\mathbf{k}} \, \hat{b}_{\mathbf{k}_p-\mathbf{k}}^\dagger(t') \right) e^{-i\mathbf{k}\cdot\hat{\mathbf{x}}(t')}. \tag{19}$$

Using (19) in (17) we obtain

$$i\hbar\frac{d}{dt}\hat{b}_{\mathbf{k}_p+\mathbf{k}} = \left( E_{\mathrm{b}} + \frac{\hbar^2\mathbf{k}^2}{2M} + \frac{\hbar\mathbf{k}\cdot\hat{\Pi}}{M} - \frac{\hbar\mathbf{k}.\hat{\mathbf{p}}}{M} \right) \hat{b}_{\mathbf{k}_p+\mathbf{k}} + V_{\mathbf{k}} - i\int_{-\infty}^\infty dt' \Gamma_{\mathbf{k}}^{(1)}(t-t') \, \hat{b}_{\mathbf{k}_p+\mathbf{k}}(t')$$

$$- i\int_{-\infty}^\infty dt' \Gamma_{\mathbf{k}}^{(2)}(t-t') \, \hat{b}_{\mathbf{k}_p-\mathbf{k}}^\dagger(t') + F_{\mathbf{k}}^{\mathrm{sto}} \tag{20}$$

where

$$\Gamma_{\mathbf{k}}^{(1)}(t-t') = \Theta(t-t') \int dq \Big( |\kappa_{q,\mathbf{k}}|^2 u_{\mathbf{k}}^2 e^{-i\omega_{q,\mathbf{k}}(t-t')} - |\kappa_{q,-\mathbf{k}}|^2 v_{\mathbf{k}}^2 e^{i\omega_{q,-\mathbf{k}}(t-t')} \Big) e^{-i\mathbf{k}(\hat{\mathbf{x}}(t')-\hat{\mathbf{x}}(t))} \tag{21a}$$

$$\Gamma_{\mathbf{k}}^{(2)}(t-t') = \Theta(t-t') \int dq \, u_{\mathbf{k}} v_{\mathbf{k}} \Big( |\kappa_{q,\mathbf{k}}|^2 e^{-i\omega_{q,\mathbf{k}}(t-t')} - |\kappa_{q,-\mathbf{k}}|^2 e^{i\omega_{q,-\mathbf{k}}(t-t')} \Big) e^{-i\mathbf{k}(\hat{\mathbf{x}}(t')-\hat{\mathbf{x}}(t))} \quad \text{and} \tag{21b}$$

$$F_{\mathbf{k}}^{\mathrm{sto}} = \int dq \left( \kappa_{q,\mathbf{k}} u_{\mathbf{k}} e^{-i\omega_{q,\mathbf{k}}t} \, \hat{\alpha}_{q,\mathbf{k}} - \kappa_{q,-\mathbf{k}}^* v_{\mathbf{k}} e^{i\omega_{q,-\mathbf{k}}t} \, \hat{\alpha}_{q,\mathbf{k}}^\dagger \right). \tag{21c}$$

Here, $\Gamma_{\mathbf{k}}^{(1)}$ and $\Gamma_{\mathbf{k}}^{(2)}$ are the memory kernels of the integro-differential equation (20), and $F_{\mathbf{k}}^{\mathrm{sto}}$ describes stochastic fluctuations due to the coupling with the bath. This will not enter in the calculation of the average trajectories and henceforth will be neglected. We next employ the Markovian approximation, which assumes no memory of past times, in (20) and under this approximation the integral kernels $\Gamma_{\mathbf{k}}^{(1)} \approx \Gamma_{\mathbf{k}}\delta(t'-t)$ and $\Gamma_{\mathbf{k}}^{(2)} \to 0$. The value $2\pi\hbar/\Gamma_{\mathbf{k}}$ gives the radiative lifetime of the polaritons. In the following we will assume

that $\Gamma_{\mathbf{k}} = \Gamma$ is a constant in $\mathbf{k}$, since the coupling constant $\kappa_{q,\mathbf{k}}$ can be well approximated as a constant for the range of wavevectors of interest in polariton experiments.

Hence, the equation of motion for $\hat{b}_{\mathbf{k}_p+\mathbf{k}}$ is simplified as

$$i\hbar \frac{d}{dt}\hat{b}_{\mathbf{k}_p+\mathbf{k}} = \left(E_{\mathrm{b}} + \frac{\hbar^2 \mathbf{k}^2}{2M} - \frac{\hbar \mathbf{k} \cdot \left(\hat{\mathbf{p}} - \hat{\Pi}\right)}{M} - i\Gamma\right)\hat{b}_{\mathbf{k}_p+\mathbf{k}} + V_{\mathbf{k}}. \tag{22}$$

The steady-state solution of (22) in the mean-field approximation is a coherent state of the Bogoliubov excitations. This follows from the fact that there are no interactions between the excitations within the Bogoliubov approximation. Under this assumption we replace the $\hat{b}_{\mathbf{k}_p+\mathbf{k}}$ quantum operator with a classical mean field $\beta_{\mathbf{k}_p+\mathbf{k}} = \langle \hat{b}_{\mathbf{k}_p+\mathbf{k}} \rangle$ in (22) and obtain the stationary solution of the field as

$$\beta_{\mathbf{k}_p+\mathbf{k}} = \frac{-V_{\mathbf{k}}}{\left(E_{\mathrm{b}} + \frac{\hbar^2 \mathbf{k}^2}{2M} - \frac{\hbar \mathbf{k} \cdot (\mathbf{p}-\Pi)}{M} - i\Gamma\right)} \tag{23}$$

Similarly the stationary solution for $\bar{\beta}_{\mathbf{k}_p+\mathbf{k}} = \langle \hat{b}^\dagger_{\mathbf{k}_p+\mathbf{k}} \rangle$ reads

$$\bar{\beta}_{\mathbf{k}_p+\mathbf{k}} = \frac{-V_{\mathbf{k}}}{\left(E_{\mathrm{b}} + \frac{\hbar^2 \mathbf{k}^2}{2M} - \frac{\hbar \mathbf{k} \cdot (\mathbf{p}-\Pi)}{M} + i\Gamma\right)}. \tag{24}$$

Notice that $\bar{\beta}_{\mathbf{k}_p+\mathbf{k}} \neq \beta^*_{\mathbf{k}_p+\mathbf{k}}$ if $E_b$ takes imaginary values. A detailed discussion of this regime is provided in Appendix A.

## 3.2 Effective mass of polarons

We next proceed in determining the observables characterizing the polaron in a driven-dissipative fluid of light. In the mean field approximation $\hat{\Pi}$ reads

$$\Pi = \sum_{\mathbf{k}\neq 0} \hbar(\mathbf{k}+\mathbf{k}_p)\bar{\beta}_{\mathbf{k}_p+\mathbf{k}}\beta_{\mathbf{k}_p+\mathbf{k}} - \hbar\mathbf{k}_p \sum_{\mathbf{k}\neq 0} \bar{\beta}_{\mathbf{k}_p+\mathbf{k}}\beta_{\mathbf{k}_p+\mathbf{k}} \tag{25}$$

The first term on the r.h.s. of (25) describes the Bogoliubov excitations mean field momentum, while the second term describes an effective drift of the impurity due to the flow of polariton fluid. We derive a self-consistent equation for $\Pi$ by substituting Eqs. (23) and (24) in (25), leading to

$$\Pi = \sum_{\mathbf{k}\neq 0} \frac{\hbar \mathbf{k} V_{\mathbf{k}}^2}{\left(E_{\mathrm{b}} + \frac{\hbar^2 \mathbf{k}^2}{2M} - \frac{\hbar \mathbf{k} \cdot (\mathbf{p}-\Pi)}{M}\right)^2 + \Gamma^2}. \tag{26}$$

In the case when $p \to 0$ the excitation momentum can be approximated as having a linear dependence in $\mathbf{p}$ coming from the first term in its Taylor expansion around $\mathbf{p} = 0$, corresponding to $\Pi \approx \eta\mathbf{p}$. In this case an effective mass of the polaron can be defined as

$$\frac{\mathbf{p}}{M_{\mathrm{eff}}} = \frac{\mathbf{p}(1-\eta)}{M} \tag{27}$$

where is $\eta$ is the fraction of excitation momentum in the direction of $\mathbf{p}$.

## 3.3 Drag force

Drag force estimates the response of a quantum fluid to a moving obstacle [78]. In particular, the vanishing of drag force provides an indication of superfluidity. The drag force experienced by an impurity is obtained, using Ehrenfest theorem, as the gradient of the impurity-fluid interaction potential:

$$\mathbf{F}_{\mathrm{drag}} = -\int d\mathbf{x} \langle \psi^\dagger(\mathbf{x}) \nabla \left[ g_{\mathrm{IB}} \, \hat{n}_{\mathrm{imp}}(x) \right] \psi(\mathbf{x}) \rangle \tag{28}$$

where $n_{\mathrm{imp}}(x) = \delta(\mathbf{x} - \hat{\mathbf{x}})$ is the density operator for a single particle (impurity). Using the Bogoliubov approximation for the fluid we obtain

$$\mathbf{F}_{\mathrm{drag}} = -\sum_{\mathbf{k}\neq 0} i\mathbf{k} V_{\mathbf{k}} \left\langle e^{i\mathbf{k}\cdot\hat{\mathbf{x}}} \left( \hat{b}_{\mathbf{k}_p+\mathbf{k}} + \hat{b}^\dagger_{\mathbf{k}_p-\mathbf{k}} \right) \right\rangle . \tag{29}$$

Using LLP transformation in the above definition, i.e. moving to the polaron reference frame, we calculate the instantaneous drag force that it experiences, which results in

$$\mathbf{F}_{\mathrm{drag}} = -\sum_{\mathbf{k}\neq 0} i\mathbf{k} V_{\mathbf{k}} \left( \beta_{\mathbf{k}_p+\mathbf{k}} + \bar{\beta}_{\mathbf{k}_p-\mathbf{k}} \right) \tag{30}$$

Substituting for $\beta_{\mathbf{k}_p+\mathbf{k}}$ and $\bar{\beta}_{\mathbf{k}_p-\mathbf{k}}$ from Eqs. (23) and (24), respectively, we have

$$\mathbf{F}_{\mathrm{drag}} = \sum_{\mathbf{k}\neq 0} 2i\,\mathbf{k} V_{\mathbf{k}}^2 \left( \frac{\hbar^2\mathbf{k}^2}{2M} + \hbar\omega_{\mathbf{k}} \right) \left( \frac{1}{\left( \frac{\hbar^2\mathbf{k}^2}{2M} + \hbar\omega_{\mathbf{k}} \right)^2 - \left( \frac{\hbar^2\mathbf{k}\cdot\mathbf{k}_p}{m} - \frac{\hbar\mathbf{k}\cdot(\mathbf{p}-\Pi)}{M} - i\Gamma \right)^2} \right) . \tag{31}$$

In the limit when $M \to \infty$ we recover the expression derived in [22].

## 3.4 Polaron trajectory

We begin deriving an equation for the polaron trajectory, by substituting Eq. (19) in (16), that is, by tracing out the bath degrees of freedom. Within the Markovian and semiclassical approximations we get

$$\frac{d\hat{\mathbf{p}}}{dt} = -\sum_{\mathbf{k}\neq 0} 2\mathbf{k} \left[ u_{\mathbf{k}}^2 \left\langle \hat{b}_{\mathbf{k}_p+\mathbf{k}}(t) \hat{b}^\dagger_{\mathbf{k}_p+\mathbf{k}}(t) \right\rangle + v_{\mathbf{k}}^2 \left\langle \hat{b}^\dagger_{\mathbf{k}_p-\mathbf{k}}(t) \hat{b}_{\mathbf{k}_p-\mathbf{k}}(t) \right\rangle \right.$$

$$\left. - u_{\mathbf{k}} v_{\mathbf{k}} \left\langle \hat{b}_{\mathbf{k}_p+\mathbf{k}}(t) \hat{b}_{\mathbf{k}_p-\mathbf{k}}(t) \right\rangle - u_{\mathbf{k}} v_{\mathbf{k}} \left\langle \hat{b}^\dagger_{\mathbf{k}_p+\mathbf{k}}(t) \hat{b}^\dagger_{\mathbf{k}_p-\mathbf{k}}(t) \right\rangle \right] . \tag{32}$$

We then use the solution of Eq. (22) for Bogoliubov operators to finally obtain a set of coupled semiclassical equations providing the trajectory of the polaron :

$$\frac{d\mathbf{x}}{dt} = \frac{(\mathbf{p}-\Pi)}{M} \tag{33}$$

$$\frac{d\mathbf{p}}{dt} = -2\Gamma \sum_{\mathbf{k}\neq 0} \frac{\mathbf{k} V_{\mathbf{k}}^2}{\left( E_{\mathrm{b}} + \frac{\hbar^2\mathbf{k}^2}{2M} - \frac{\hbar\mathbf{k}\cdot(\mathbf{p}-\Pi)}{M} \right)^2 + \Gamma^2}$$

$$\left\{ 1 - 2\cos\left[ \left( E_{\mathrm{b}} + \frac{\hbar^2\mathbf{k}^2}{2M} + \frac{\hbar\mathbf{k}\cdot(\Pi-\mathbf{p})}{M} \right) \frac{t}{\hbar} \right] e^{-\frac{\Gamma t}{\hbar}} + e^{-\frac{2\Gamma t}{\hbar}} \right\} . \tag{34}$$

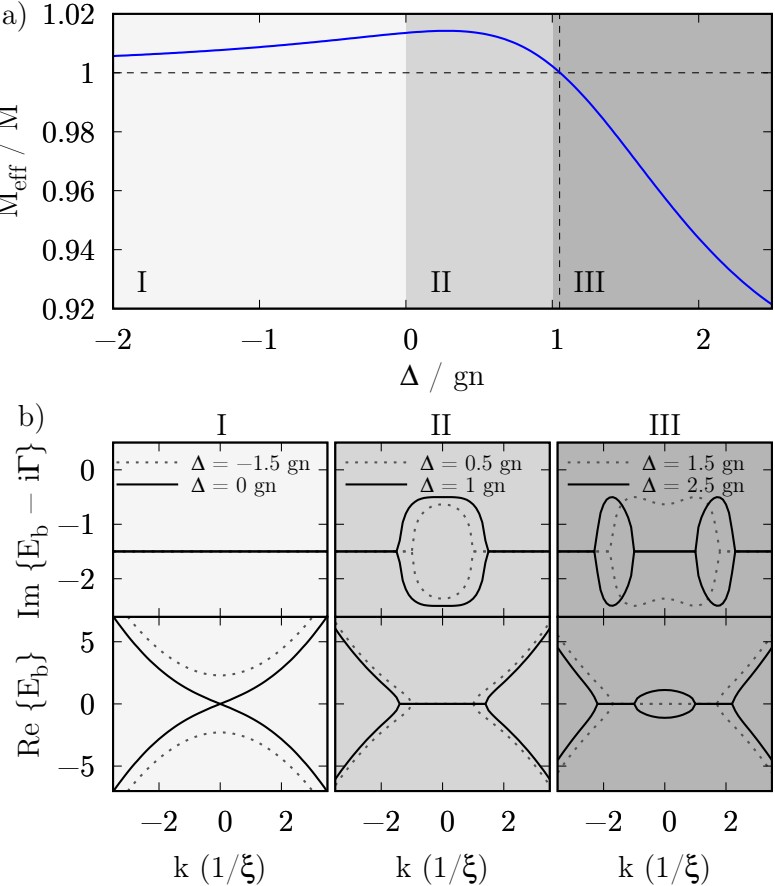

Figure 2: **a) Ratio of effective mass to bare mass,** $M_{\text{eff}}/M$ of the impurity as a function of the detuning, $\Delta$ - in the units of blueshift $gn$. The shaded regions from left to right depicts the regions (I), (II) and (III) corresponding to different types of excitation spectrum of the fluid shown in Panel b) and the dashed lines indicate the point where effective mass coincides with bare mass of the impurity. Here the impurity-polariton mass ratio is $M/m = 10$. **b) Imaginary (top) and real (bottom) parts of the Bogoliubov spectrum of polaritons** for varying values of detuning $\Delta$ leading to (I) gapless or gapped spectrum, (II) diffusive spectrum around $\mathbf{k} = 0$ and (III) diffusive spectrum around finite $\mathbf{k}$ points. In all the figures we have used $\Gamma = 1.5\,gn$ and $\mathbf{k}_p = 0$.

# 4 Results

## 4.1 Effective mass of the polaron

We present, first, the results for the polaron effective mass. They are obtained by using the numerically evaluated self-consistent solution of Eq. (26) at small values of impurity momentum $p$, from which the effective mass is extracted according to its definition in Eq. (27)[1]. In the current and following subsections we present our results in a quasi-one dimensional geometry, as illustrated in Fig.1, but the equations that we derived are also valid in higher dimensions. Also, for all further results we will take the value of interaction constant, $(1/2\pi n\xi)(g_{\mathrm{IB}}/g)^2$, with $\xi = \hbar/\sqrt{mgn}$ being the healing length, equal to 0.2. This ensures we stay in the regime of weak impurity-fluid interactions, as assumed in Sec.2.

As shown in Figure 2a), upon increasing the detuning $\Delta$ we observe a non-monotonous behaviour of the effective mass. This can be understood as being related to the different nature of the Bogoliubov excitations at varying $\Delta$. The various possible regimes are illustrated in Fig. 2b). When $\Delta \leq 0$ (region I in the figure) the excitation spectrum is gapped and becomes gapless and phononic at $\Delta = 0$. The latter regime is reminiscent of the Bogoliubov spectrum for equilibrium condensates albeit with dissipation. When $\Delta > 0$ the excitation spectrum is diffusive with purely imaginary eigenenergies in some range of wavevectors that depends on the ratio $\Delta/gn$. If $\Delta/gn < 1$ (region II) the imaginary values lie in the region $|\mathbf{k}| < \sqrt{2m\Delta}$ with the maximum of the imaginary part at $\mathbf{k} = 0$, while if $\Delta/gn > 1$ (region III) the diffusive regime is at finite $\mathbf{k}$ wavevectors.

In all the above cases, the polariton condensate is dynamically stable as long as the imaginary part of its excitation spectrum is smaller than zero [9,79], since the presence of a positive imaginary part leads to a dynamical instability, occurring when $\mathrm{Im}[E_{\mathrm{b}}] > 0$. This condition corresponds to an exponential growth in time of the population of excitation modes and yields a rapid depletion of the polariton condensate. The presence of dissipation protects the system against such instabilities as long as $\Gamma > \mathrm{Im}[E_{\mathrm{b}}]$. The failure to meet this condition leads to the breakdown of Bogoliubov theory, and our theory is not valid for such instances (as depicted by the shaded gray area in Figure 5). Hence, it is remarkable that driven-dissipative quantum fluids can reach a wealth of dynamical regimes not accessible by their equilibrium counterpart. This has a direct impact on the polaron effective mass: the way the fluid excitations dress the impurity depends on the dynamical regime. We find that the largest effective mass occurs close to the sonic case, where a large density of excitations can be generated by the impurity for vanishing energetic costs. Another peculiar regime found in this driven-dissipative quantum fluids is the one where the effective mass is smaller than the bare one: this occurs when the spectrum is diffusive at finite wavevectors. In this case the impurity feels a negative drag (see next subsection) and behaves as an effectively lighter particle.

## 4.2 Drag force

We next calculate the drag force experienced by the polaron by numerical integration of Eq. (31) in one dimension. The drag force depends on the fluid momentum $\hbar k_p$, and on the dynamical regime for the fluid excitations, controlled by the detuning $\Delta$. The results are summarized in Fig. 3.

For positive values of $\Delta$ we observe a region of negative drag force. This extends the results predicted by [22] to the case of an impurity with finite mass. This regime

---

[1]For the case of diffusive spetrum, $\Delta > 0$, the effective mass was extracted using the real part of the impurity momentum $\Pi$, as its imaginary part adds to the dissipation of the polariton fluid hence does not contribute to the effective mass.

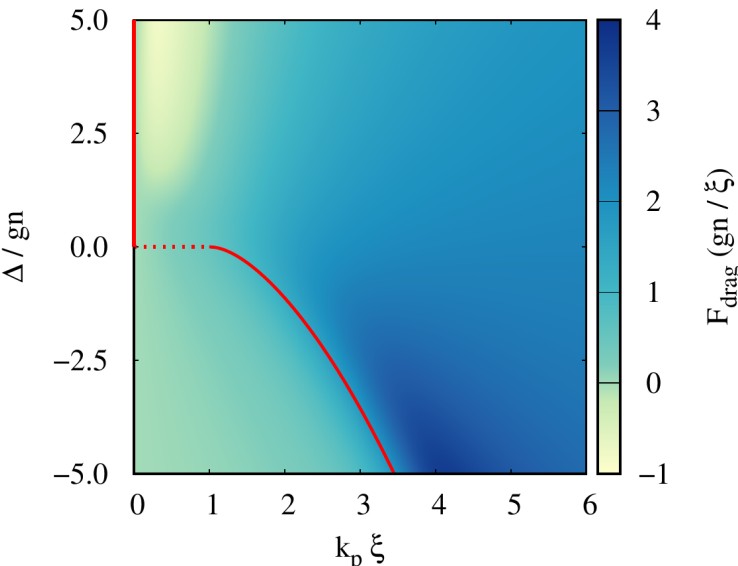

Figure 3: **Drag Force**, $F_{\mathrm{drag}}$, experienced by polaron at rest as a function of the detuning $\Delta$ - in units of interation energy, $gn$ - and the polariton pump wavevector $\mathbf{k}_p$ - in units of inverse healing length $\xi = \hbar/\sqrt{mgn}$ - which quantifies the fluid flow velocity, $v_{\mathrm{fl}} = \hbar\mathbf{k}_p/m$. The red line depicts the Landau critical velocity, $v_c$ as function of detuning $\Delta$, with $v_c = \min_{\mathbf{k}} E_b(\mathbf{k})/|\mathbf{k}|$. Here we have taken $M/m = 10$ for the impurity-polariton mass ratio and $\Gamma = 1.5\,gn$.

corresponds to the case when the effective mass is smaller than bare one. For negative values of $\Delta$ the drag force is non-zero and positive, showing an important increase starting from the case where the fluid velocity exceeds the Landau critical velocity, estimated as $v_c = \min_{\mathbf{k}} E_b(\mathbf{k})/|\mathbf{k}|$ and marked by the red line in Fig. 3. Our analysis hence shows that the impurity acts as a 'test particle' in the fluid to probe its superfluid properties, which strongly depend on the dynamical regime of the fluid.

## 4.3  Polaron dynamics

We finally follow the semiclassical dynamics of the polaron moving with a finite initial momentum in a fluid at rest, obtained by numerically solving the coupled differential equations in Eqs. (33) and (34). The results are summarized in Fig. 4.

The key quantity to follow in order to determine the impurity dynamics is the excitation momentum $\Pi$. When $\Pi$ reaches zero, it means that the impurity has reached its terminal velocity: it is not dressed by excitations anymore, and its effective mass equals the bare mass. Different kind of trajectories are shown in Fig. 4 a). The arrows indicate the impurity momentum evolution in time, till it reaches a terminal value (black dots in the figure). Note that at long times, we have checked that the relation $dp/dt = -F_{drag}(p)$ is valid and helps understand these trajectories. For negative $\Delta$ the drag force is positive, the impurity decelerates and the terminal momentum is always zero regardless of the initial momentum. For $\Delta > 0$, the situation is more exotic: above a certain positive $\Delta$, the terminal momentum $p = 0$ becomes unstable with respect to fluctuations of $\Pi$, and two nonzero terminal momenta of opposite sign become possible (cf. the bifurcation in the inset of Fig. 4 a)). These nonzero terminal momenta result from the negative drag regime in which the impurity is accelerated. Note that this trajectory does not violate energy conservation in this driven-dissipative situation: the energy flux constantly traversing

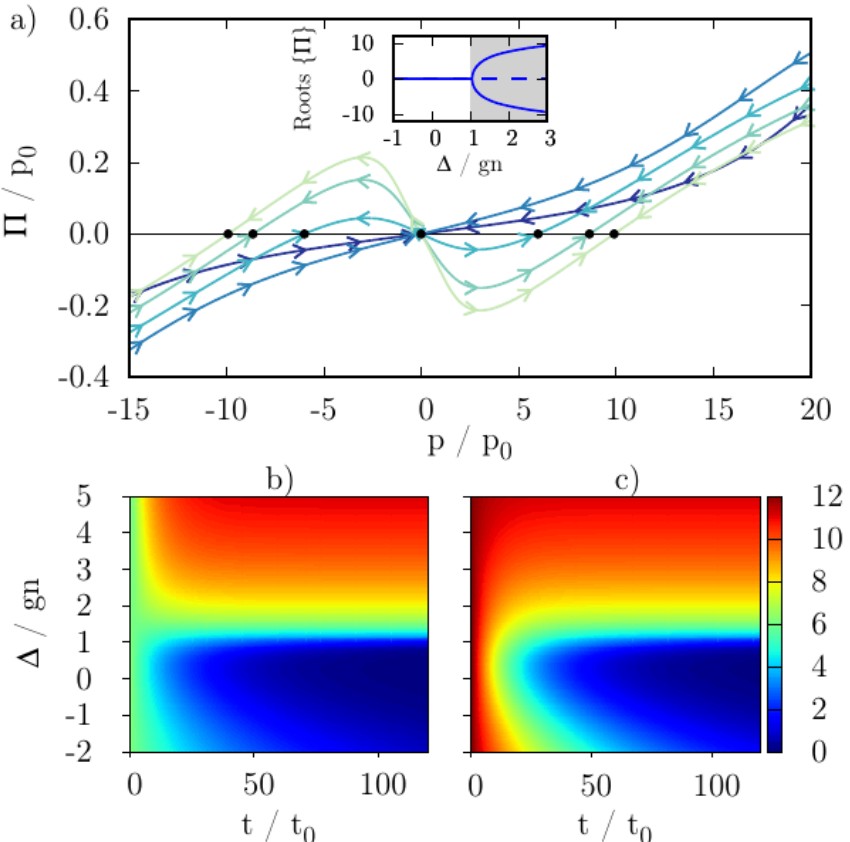

Figure 4: **a) Excitation momentum** $\Pi$ as a function of polaron momentum $p$, both in units of $p_0 = \sqrt{mgn}$ for various values of detuning $\Delta = -1.5, 0, 1.5, 2.5, 3.5$ - in units of $gn$ - going from darker to lighter shaded curves respectively, for a fluid at rest. The arrows depict the momentum flow towards the long time limit state, where the polaron attains a saturation or terminal momentum. The saturation momentum values are depicted by the black dots on $p$ axis (roots of excitation momentum $\Pi(p)$) and are dependent on the initial value of the impurity momentum $p$. In the inset we show the roots of the excitation momentum, as a function of $\Delta$ in units of $gn$. The gray-shaded region in the inset marks the region of negative drag. **b) and c) Heatmaps of impurity momentum**, $\mathbf{p}/p_0$ as a function of time, $t$, in units of $t_0 = \hbar/gn$, and detuning $\Delta$, in units of $gn$, for different values of the inital momentum: c) $p = 6\,p_0$, d) $p = 12\,p_0$. In all panels we have taken $M/m = 10$, $\Gamma = 1.5\,gn$ and the fluid is at rest.

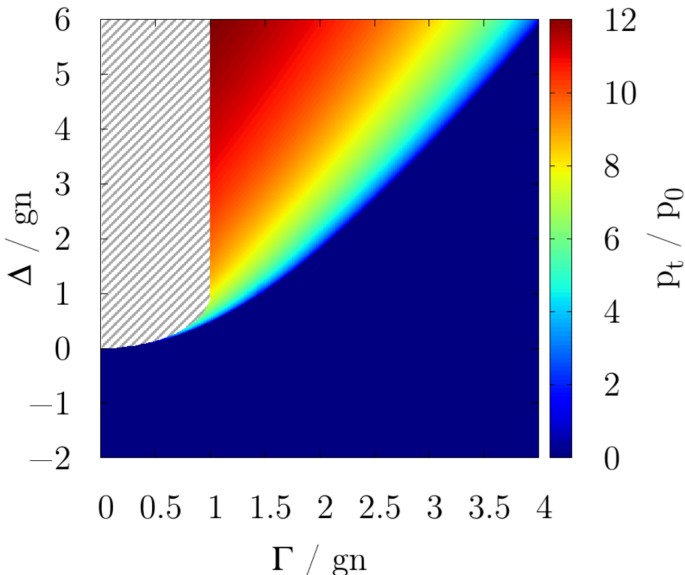

Figure 5: **Polaron saturation momentum**: absolute value of the polaron saturation momentum $p$ in units of $p_0 = \sqrt{mgn}$ as a function of detuning, $\Delta$, and dissipation, $\Gamma$ - both in units of blueshift $gn$, for a fluid at rest. The gray shaded area represents the region of dynamical instability of the Bogoliubov theory. The region separating zero and positive saturation momentum of the impurity corresponds to the negative drag regime.

the polariton fluid (constituted by the drive and losses) provides the energy that such trajectories require.

In Figure 4b) and Figure 4c) we show the polaron momentum as a function of time $t$ and detuning $\Delta$ for fixed initial momentum, smaller or larger than the terminal value. Two regimes clearly emerge from this analysis: for negative detuning the polaron decelerates till a final rest position, while for positive detuning, in the regime of negative drag, the impurity of given initial momentum accelerates till a terminal momentum is reached.

Finally, we analyse the influence of losses on the polaron dynamics for positive $\Delta$. The results are summarized in Fig. 5, where we show the terminal velocity reached by polarons as a function of the detuning $\Delta$ and of the dissipation constant $\Gamma$. We see that the region of non-zero terminal velocity caused by the negative drag regime disappears upon increasing $\Gamma$. For increasing $\Gamma$ we also see that the lower boundary of this region increases to compensate for the increased losses.

## 5  Conclusions and outlook

We have studied the motion of an impurity in a polariton fluid under drive and dissipation, assuming a weak coupling between the impurity and the fluid. The presence of Bogoliubov excitations lying on top of the coherent steady state of the polariton fluid dress the impurity particle giving rise to a Bose polaron in the Fröhlich regime. We have determined the polaron effective mass, the drag force acting on the impurity, as well as polaron trajectories at semiclassical level.

We have found different dynamical regimes, originating from the unique features of the excitation spectrum of driven-dissipative polariton fluids. We have shown that it is possible to tune polaron effective mass to values both smaller and larger than the bare one by adjusting the detuning $\Delta$. In the $\Delta > 0$ regime of diffusive excitation spectrum,

for specific $\Delta$ values corresponding to the case of finite terminal momentum in Fig. 5, the impurity is subjected to an instantaneous negative drag force: as a result, the impurity rest position is unstable and counter-intuitively, it starts accelerating against the flow until it reaches a non-zero terminal velocity. This work shows that the impurity dynamics can be used as a test particle to probe the different regimes of nonequilibrium quantum flow, including superfluidity, in quantum fluids of light.

As indicated by our analysis of the coupling with the electromagnetic vacuum bath outside the cavity, corrections beyond the Markov approximation (see [80] for an approximate treatment) could lead to experimentally relevant non-trivial corrections of the dynamics, that would be interesting to examine. Another open direction is to go beyond the semiclassical description of the impurity trajectories, and beyond the weak impurity-fluid interaction regime.

# Acknowledgements

**Funding information**    AV acknowledges the European Union Horizon 2020 research and innovation program under the Marie Skłodowska-Curie Grant Agreement No. 754303.

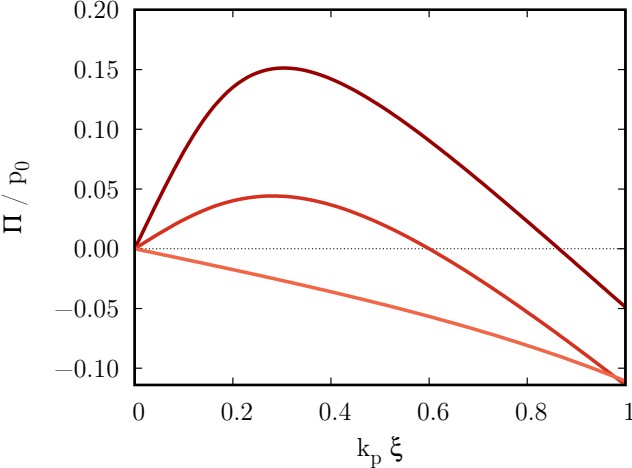

Figure 6: **Finite flow velocity**: excitation momentum, $\Pi$ in units of $p_0 = \sqrt{mgn}$ as a function of the pump wavector $k_p$, in units of inverse healing length $\xi$ for an impurity at rest. Here the values of detuning, $\Delta = 2.5, 1.5$ and $-1$, decrease with the color from a darker to a lighter shade and we have taken $M/m = 10$ and $\Gamma = 1.5\, gn$.

# A    Generalized Bogoliubov transformation

In the case of positive detuning, the quadratic form of lower polariton Hamiltonian in (5) after undergoing Bogoliubov approximation is non-positive definite for a finite range of wavevector $\mathbf{k}$ values. This precludes its diagonalization using the standard Bogoliubov operators $\hat{b}_{\mathbf{k}}$ and $\hat{b}_{\mathbf{k}}^{\dagger}$ (6a, 6b) in this range of $\mathbf{k}$ values [81]. This happens because $\hat{b}_{\mathbf{k}}$ and $\hat{b}_{\mathbf{k}}^{\dagger}$ fail to satisfy the bosonic commutation relations upheld by the condition $|u_{\mathbf{k}}|^2 - |v_{\mathbf{k}}|^2 = 1$ which in this instance is zero. However, we may still diagonalize the Hamiltonian using

non-standard Bogoliubov operators given by

$$\hat{b}_{\mathbf{k}_p+\mathbf{k}} = u_{\mathbf{k}}\,\hat{a}_{\mathbf{k}_p+\mathbf{k}} + v_{-\mathbf{k}}\,\hat{a}^\dagger_{\mathbf{k}_p-\mathbf{k}} \tag{35}$$

$$\hat{\bar{b}}_{\mathbf{k}_p-\mathbf{k}} = v_{\mathbf{k}}\,\hat{a}_{\mathbf{k}_p+\mathbf{k}} + u_{-\mathbf{k}}\,\hat{a}^\dagger_{\mathbf{k}_p-\mathbf{k}}. \tag{36}$$

These operators satisfy the bosonic commutation relation with the condition $u_{\mathbf{k}}^2 - v_{\mathbf{k}}^2 = 1$ but in this case $\hat{\bar{b}}_{\mathbf{k}_p+\mathbf{k}} \neq \hat{b}^\dagger_{\mathbf{k}_p+\mathbf{k}}$ which results, in general, complex Bogoliubov energy and onset of dynamical instability. However, we stay in the regime where the imaginary part of the Bogoliubov energy is less than the dissipation rate, $\Gamma$ which stabilizes the system against the instability caused due to complex Bogoliubov energies. Proceeding forward with these non-standard Bogoliubov operators all the subsequent results remain the same as derived in the main paper, albeit replacing $\hat{b}^\dagger_{\mathbf{k}_p+\mathbf{k}}$ with $\hat{\bar{b}}_{\mathbf{k}_p+\mathbf{k}}$.

The use of non-standard Bogoliubov operators when detuning $\Delta$ is positive also results in a small imaginary contribution to the excitation momentum $\Pi$ when we solve the self-consistency relation in (26). We have taken it into account by adding an imaginary contribution to the energy spectrum as $\mathrm{Im}\{E_{\mathrm{b}} - i\Gamma\} + \mathrm{Im}\{\Pi\}$.

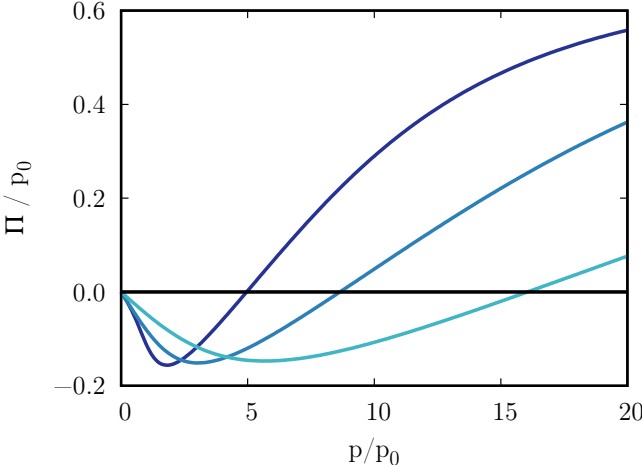

Figure 7: **Varying impurity-polariton mass ratio**. Excitation momentum, $\Pi$ as a function of impurity momentum $p$, both in units of $p_0 = \sqrt{mgn}$ in the case of diffusive Bogoliubov spectrum ($\Delta = 2.5\,gn$) for varying impurity-polariton mass ratios $M/m$: 5, 10 and 20 from a darker to a lighter shade respectively. The case for $M/m = \infty$ is shown in black for which the excitation momentum is zero. Here $\Gamma = 1.5\,gn$ and the fluid is at rest.

# B   Further Results

## B.1   Case of moving fluid

The case of a moving fluid with a finite flow velocity $|\mathbf{v}| = k_p/m$ is a straightforward extension of the case discussed in the main text. We focus here on the regime of diffusive spectrum. As shown in Fig. 6, the terminal drift velocity attained by the impurity depends on the detuning $\Delta$. For sufficiently large $\Delta$ the regime of negative drag emerges, indicated by a terminal velocity in the opposite direction of fluid flow.

## B.2 Varying mass ratios

We show in Fig. 7 the results for excitation momentum $\Pi$ in the case of diffusive spectrum for varying values of the impurity-polariton mass ratios, $M/m$. We observe that for increasing mass ratio the terminal velocity attained by the impurity also increases.

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
