# Peer review of "Bose polaron in a quantum fluid of light"

_SciPost Physics_

## Round 1 · Referee Report · Anonymous · 2021-8-25

Report

The manuscript by A. Vashisht et al. discusses the properties of a single impurity in a driven-dissipative polariton bath. The paper is very well written, the problem is experimentally interesting, the formalism is clearly explained, and the equations and the results appear correct. As such, I can surely recommend publication of this manuscript in SciPost.

Upon resubmission, I think the Authors may extend the discussion on the key result presented here, i.e., the presence of a region of negative effective mass. In particular, is there an intuitive understanding of why such effect appears when the Bogolubov dispersion becomes complex? If I understood correctly, such effect was already noticed in earlier works on static impurities (Refs. [22,23]). In this case, this discussion may be placed already in the Introduction.

Here below follow further minor remarks and suggestions, presented in chronological order, that the authors may want to take into account.

# caption of Fig. 1: "Central panel as a" ---> "Central panel: as a " (":" missing). Moreover, define what is an "LP fluid".

# Sec. 2, line 3: "by its coupling" ---> "by their coupling"

# one line below Eq. (1): explain the adjective "lower" used in front of "polariton"

# Eqs. (2) and (4): define the operators $a_k$ and $\alpha_k$

# Eq. (5): in the moving reference frame, shouldn't the two operators $a_{k_p}$ and $a^\dagger_{k_p}$ in the last line of the equation actually read $a_0$ and $a^\dagger_0$?

# one line below Eq. (9): "all the the" ---> "all the". Further, the discussion in this paragraph is unclear. It is first said that $u_k = u^*_k$, but then it is specified that when $\Delta>0$ this is not the case. To avoid ambiguities, I suggest then to say "if $\Delta>0$ then $u_k = u^*_k$. Else, ..."

# to simplify notation, I suggest to introduce the operator $\Pi$ already around Eq. (13).

# Eqs. (17) and (20): a factor $1/M$ is missing in the term $\hbar k \cdot p$

# two lines above Eq. (22): $2\pi/\Gamma_k$ ---> $2\pi/\Gamma_{\bf k}$

# Eq. (28): $\langle \int$ ---> $\int \langle$. Moreover, I have the impression that $n_{imp}$ is not defined.

# line following Eq. (29): "ie" ---> "i.e."

# Eq. (34): I have the impression that $\Gamma$ (without pedix k) is not defined, please check. It is also unclear how such quantity emerges from the previous equations. A few more lines of explanation may be spent to explain this key equation.

# end of page 9: The two sentences "In all the above cases... . This is possible ..." contain important information, which I think deserves further clarification for the reader unfamiliar with polaritons. The authors may want to extend slightly the discussion here.

# Fig.2b: inside the various panels, $\Delta = ...$ should be $\Delta/gn = ...$

# Fig. 3: what is the dashed part of the red line?

# last paragraph of Sec. 4: "the region ... disappear" ---> "the region ... disappearS"

---

## Round 1 · Referee Report · Anonymous · 2021-9-15

Strengths

1-Detailed theoretical description
2-Effective graphics
3-Well written

Weaknesses

1-Solution methods are not stated
2-Lacks some discussion

Report

This work reports a study of an impurity in a non-equilibrium exciton-polariton condensate, with the dynamics of both the quantum fluid and the impurity taken into account. A peculiar regime of negative drag on the impurity is identified and characterized. The paper is well written and the theoretical description is clearly presented. The results are interesting, seem sound and are nicely illustrated by the figures. I think however that some additions are needed to make the paper clearer and more relevant.

First of all I think that the means (numerical, analytical...) with which the results are obtained should be stated more clearly, for example at the beginning of Section 4.
Another point that in my opinion is important to discuss is the experimental relevance of these calculations. In particular: how would a moving impurity be realized in an exciton-polariton condensate?
Moreover I think that the results of Ref. [22] are not quoted with sufficient clarity, since some of the results obtained here regarding the negative drag were anticipated in that paper, albeit with a much simpler model of a fixed impurity. A discussion of those results should be included in the Introduction.
Finally the exposition may benefit from adding some explanations to the theoretical part of Section 3, as I indicate in the "Requested changes", where I also indicate some minor points.

Even if I think that (after these changes) this work is well worth publishing, I am not sure that it meets one of the very selective expectations of SciPost Physics. While the paper contains interesting results obtained in an original way, I must admit that I do not see a groundbreaking discovery, a novel link between different fields or much follow-up work. For these reasons in my opinion the paper would fit better in SciPost Physics Core.

Requested changes

In order of appearance
1-I think the predictions of Ref.[22] should be reported in the introduction
2-A discussion of the experimental relevance of the calculations should be included
3-Operators $\hat{a}_\mathbf{k}$ in Eq. (2) and $\hat{\alpha}_\mathbf{k}$ in Eq. (4) are not defined
4-Maybe the $\hat{p}$ operator in equation (3) (and also in (10)) should be bold
5-I think the density n appearing after equation (8) is not defined
6-At the end of Sec. 2.1 and at the beginning of Sec. 2.2 it would be nice to have references for the Bogoliubov-Frolich Hamiltonian and the Lee Low Pines transformation
7-When $\Pi$ is an operator it should have the hat
8-After Eq. (22) a coherent state of excitations is considered. When is this a valid procedure? I think some comment is worth adding.
9-At the end Sec. 3.2 I think the definition of the effective mass and of $\eta$ and how they follow from Eq. (26) should be explained in more detail
10-Also the definition of the drag force in Eq. (28) would benefit from some more details. Also I think $n_{imp}$ is not defined.
11-Eq. (28) and (29) should have only one number
12-In the line after (29) ie should be i.e.
13-A discussion of the solution methods should be added in Section 4
14-At the end of the second paragraph of Sec. 4.1 the value $\Delta/gn=2$ is used to distinguish the different regimes, but in Fig. 2 $\Delta/gn=1$ appears as a threshold
15-I could not find a definition of the $\Gamma$ independent of $\mathbf{k}$ that is used in Section 4
16-Is "decreasing $\Gamma$" in at the end of Sec. 4 supposed to be "increasing $\Gamma$"
17-In the conclusions it is stated that for $\Delta>0$ the impurity is subjected to negative drag force, but this does not always happen for $\Delta>0$
18-In the caption of Fig. 6, in the list of values of $\Delta$, there is a cursive "and"

---

## Round 2 · Referee Report · Anonymous · 2021-10-22

Report

The authors responded accurately and satisfactorily to all my comments, and in my opinion also to the ones of the other referee. As such, at this stage I can fully recommend publication of this manuscript in Scipost Physics.

---

## Round 2 · Referee Report · Anonymous · 2021-10-26

Report

The authors satisfactorily addressed all the requested changes. I hence think that the article can be published in its present form.
However in the previous report I expressed my doubts that this article meets one of the very selective expectations of SciPost Physics and I suggested SciPost Physics Core as an alternative. The authors did not discuss this so my suggestion remains.

  • validity: -
  • significance: -
  • originality: -
  • clarity: -
  • formatting: -
  • grammar: -

Author:  Amit Vashisht  on 2021-11-02  [id 1902]

(in reply to Report 2 on 2021-10-26)
Category:
answer to question
reply to objection

We thank the Referee for his/her positive assessment of our resubmitted article. We didn't consider SciPost Physics Core as an alternative journal because, to the best of our knowledge, we are the first to address the problem of Bose Polaron in an out of equilibrium setting. The Bose Polaron problem in equilibrium quantum gases is important and largely studied. Likewise, we believe that our results will open a new research direction of interest for a broad community ranging from photonics to quantum gases.

---

## Round 2 · Author Response

Reply to Referee 1

System Message: WARNING/2 (<string>, line 1)

Title overline too short.
* * *
Reply to Referee 1
* * *
We express our gratitude to the Referee for her/his favourable review of our work and recognizing its significance. Following her/his suggestions, we have introduced a small extension of the second last paragraph in the introduction highlighting the intuition behind the negative drag force phenomenon and its link to the complex Bogoliubov spectrum. We also thank the Referee for thorough reading of our manuscript and pointing out corrections in the text, which we now have rectified. The changes are listed below in list of changes.

Reply to Referee 2

System Message: WARNING/2 (<string>, line 6)

Title overline too short.
* * *
Reply to Referee 2
* * *
We express our gratitude to the Referee for her/his valuable review of our work and for suggesting changes to increase its relevance. We have made the below changes to the manuscript in line with your suggestions.

---

## Round 2 · List of Changes

* * *
Changes suggested by Referee 1
* * *
# caption of Fig. 1: "Central panel as a" ---> "Central panel: as a " (":" missing). Moreover, define what is an "LP fluid".
> We have introduced the concept of lower polaritons in the first paragraph of Section 2. We also added the explanation on why we neglect the upper-polariton. For better readability, we have taken out the acronym LP from the figure 1 and the caption and replaced it as ‘polariton fluid’.

# Sec. 2, line 3: "by its coupling" ---> "by their coupling"
> The suggested change has been implemented.

# one line below Eq. (1): explain the adjective "lower" used in front of "polariton"
> Same as above.

# Eqs. (2) and (4): define the operators $ a_k $and $ \alpha_k $
> We have added the definition of both the operators in their respective places just below Eq.2 and Eq.4 respectively.

# Eq. (5): in the moving reference frame, shouldn't the two operators $ a_{k_p} $ and $ \alpha_{k}$ in the last line of the equation actually read $ a_0 $ and $ a_0^{\dagger} $?
> We thank the Referee for the question. An inaccuracy had slipped in the presentation; there is no change of the reference frame leading to Eq.(5), just a change of variable under the summation sign. For this reason, the two operators in the last line remain $ a_{k_p} $ and $ a^{\dagger}_{k_p} $. We have corrected the sentence above Eq.(5) accordingly.

# one line below Eq. (9): "all the the" ---> "all the". Further, the discussion in this paragraph is unclear. It is first said that $ u_k = u_k^* $ but then it is specified that when $ \Delta > 0 $ this is not the case. To avoid ambiguities, I suggest then to say "if $ \Delta > 0 $ then $ u_k = u_k^* $. Else, ..."
> We took into account the ambiguities that may arise and have rephrased the paragraph below Equation 9.

# to simplify notation, I suggest to introduce the operator $ \Pi $ already around Eq. (13).
> We have followed the suggestion and introduced the operator $ \Pi $ just below Equation 13.

# Eqs. (17) and (20): a factor $ 1/M $ is missing in the term $ \hbar k \cdot p $.
> The missing factor has been inserted in both the equations.

# two lines above Eq. (22): $ 2 pi/Gamma_k $ --->$2 pi/Gamma_{\bf k}$
> The correction has been implemented.

# Eq. (28): ⟨∫ ---> ∫⟨. Moreover, I have the impression that $ n_{\rm imp} $ is not defined.
> We have provided a definition of $ n_{\rm imp}$ for our case of impurity below Equation 28 and corrected the equation as suggested.

# line following Eq. (29): "ie" ---> "i.e."
> The correction has been implemented.

# Eq. (34): I have the impression that $ \Gamma $ (without pedix k) is not defined, please check. It is also unclear how such quantity emerges from the previous equations. A few more lines of explanation may be spent to explain this key equation.
> We have written more clearly at which point of the derivation we drop the k-dependence in Gamma and added an explanation for this approximation before Eq. (22)

# end of page 9: The two sentences "In all the above cases... . This is possible ..." contain important information, which I think deserves further clarification for the reader unfamiliar with polaritons. The authors may want to extend slightly the discussion here.
> We have followed the suggestion and extended the discussion at the beginning of the last paragraph before Section 4.2 to explain in a clearer way the criteria for stability of the polariton system within the context of Bogoliubov theory.

# Fig.2b: inside the various panels, $ \Delta = ... $ should be $ \Delta/gn = … $
> The figure has been rectified.

# Fig. 3: what is the dashed part of the red line?
> The dashed part is just a guide to eye linking the regimes with $ \Delta > 0 $ and $\Delta < 0 $. The red line is obtained from the Landau criterion, allowing us to obtain the critical kp as a function of Delta. The solution suddenly changes from finite kp to zero when the dispersion relation changes its form.

# last paragraph of Sec. 4: "the region ... disappear" ---> "the region ... disappears"
> The correction has been implemented.
* * *
Changes suggested by Referee 2
* * *
1) I think the predictions of Ref.[22] should be reported in the introduction.
We have introduced a small extension at the end of the third paragraph in the introduction stating the predictions of Ref [22].

2) A discussion of the experimental relevance of the calculations should be included.
Following the suggestion of the Referee, we now give, in the second last paragraph of the introduction, a presentation of realistic experimental strategies to create an impurity in an exciton polariton fluid.

3) Operators $ a_k $ in Eq. (2) and $ \alpha_k $ in Eq. (4) are not defined.
The definition of the two operators has been included in the relevant places just below Eq. 2 and Eq. 4 respectively.

4) Maybe the $ p $ operator in Eq. (3) (and also in (10)) should be bold
The change has been implemented.

5) I think the density n appearing after equation (8) is not defined
A definition for density n has been added one line below Equation 8

6) At the end of Sec. 2.1 and at the beginning of Sec. 2.2 it would be nice to have references for the Bogoliubov-Frolich Hamiltonian and the Lee Low Pines transformation
We have added references for both in the relevant places in Section 2.

7) When $ \Pi $ is an operator it should have the hat
The change has been implemented.

8) After Eq. (22) a coherent state of excitations is considered. When is this a valid procedure? I think some comment is worth adding.
We followed the suggestion and added a statement one line below Eq. (22). Since our Hamiltonian is assumed quadratic after the Bogoliubov approximation, we have neglected any interactions between the excitations or Bogoliubov modes, hence, the steady state for the independent Bogoliubov modes will give us a coherent state for that mode.

9) At the end Sec. 3.2 I think the definition of the effective mass and of $ \eta $ and how they follow from Eq. (26) should be explained in more detail
We have rephrased the part below Eq.(26) to explain more clearly how we can write an equation for the effective mass of the polaron in the limit when the impurity momentum tends to zero.

10) Also the definition of the drag force in Eq. (28) would benefit from some more details. Also I think $ n_{\rm imp}$ is not defined.
The definition of the impurity density has been added below Eq.(28). Following the Referee’s suggestion, we have introduced the concept of drag force.

11) Eq. (28) and (29) should have only one number
We have split the two equations for clarity of the concept of drag force as suggested above by the Referee and hence they still have two different numbers.

12) In the line after (29) ie should be i.e.
The change has been implemented.

13) A discussion of the solution methods should be added in Section 4
We have added a statement in the first paragraph of Subsections 4.1, 4.2 and 4.3 stating the solution method implemented for the calculation of results relevant to each subsection.

14) At the end of the second paragraph of Sec. 4.1 the value $ \Delta/gn = 2 $ is used to distinguish the different regimes, but in Fig. 2 $ \Delta/gn = 2 $ appears as a threshold.
Thank you for pointing out the error, it was a typographical mistake in the second paragraph of Section 4.2. The threshold is indeed at $ \Delta/gn = 1 $ and not 2 as was stated before. We have rectified that.

15) I could not find a definition of the $ \Gamma $ independent of $ k $ that is used in Section 4
We have added a definition for dissipation $ \Gamma $ independent of $ k $ three lines before Equation 22 and continued with a constant dissipation further onwards so that there is no confusion once we are analysing the results in Section 4.

16) Is "decreasing $ \Gamma $" in at the end of Sec. 4 supposed to be "increasing $ \Gamma $"
Yes, indeed. The change has been implemented.

17) In the conclusions it is stated that for $ \Delta > 0 $ the impurity is subjected to negative drag force, but this does not always happen for $ \Delta > 0 $.
We have extended the sentence to explain this point and made reference to Fig.5 to illustrate the regime of Delta for which negative drag occurs.

18) In the caption of Fig. 6, in the list of values of $ \Delta $, there is a cursive "and"
The typographical correction has been implemented.

---

## Editorial Decision

publication_decision_taken:_accept